# Renal Ischemia Tolerance Mediated by eIF5A Hypusination Inhibition Is Regulated by a Specific Modulation of the Endoplasmic Reticulum Stress

**DOI:** 10.3390/cells12030409

**Published:** 2023-01-25

**Authors:** Nicolas Melis, Isabelle Rubera, Sebastien Giraud, Marc Cougnon, Christophe Duranton, Mallorie Poet, Gisèle Jarretou, Raphaël Thuillier, Laurent Counillon, Thierry Hauet, Luc Pellerin, Michel Tauc, Didier F. Pisani

**Affiliations:** 1Laboratory of Cellular and Molecular Biology, Center for Cancer Research, National Cancer Institute, Bethesda, MD 20892, USA; 2Université Côte d’Azur, CNRS, LP2M, 06108 Nice, France; 3Laboratories of Excellence Ion Channel Science and Therapeutics, 06103 Nice, France; 4INSERM U1313, IRMETIST, Université de Poitiers et CHU de Poitiers, 86000 Poitiers, France

**Keywords:** GC7, hypusine, preconditioning, XBP1, BiP, anoxia, kidney

## Abstract

Through kidney transplantation, ischemia/reperfusion is known to induce tissular injury due to cell energy shortage, oxidative stress, and endoplasmic reticulum (ER) stress. ER stress stems from an accumulation of unfolded or misfolded proteins in the lumen of ER, resulting in the unfolded protein response (UPR). Adaptive UPR pathways can either restore protein homeostasis or can turn into a stress pathway leading to apoptosis. We have demonstrated that N1-guanyl-1,7-diamineoheptane (GC7), a specific inhibitor of eukaryotic Initiation Factor 5A (eIF5A) hypusination, confers an ischemic protection of kidney cells by tuning their metabolism and decreasing oxidative stress, but its role on ER stress was unknown. To explore this, we used kidney cells pretreated with GC7 and submitted to either warm or cold anoxia. GC7 pretreatment promoted cell survival in an anoxic environment concomitantly to an increase in *xbp1* splicing and BiP level while eiF2α phosphorylation and ATF6 nuclear level decreased. These demonstrated a specific modulation of UPR pathways. Interestingly, the pharmacological inhibition of *xbp1* splicing reversed the protective effect of GC7 against anoxia. Our results demonstrated that eIF5A hypusination inhibition modulates distinctive UPR pathways, a crucial mechanism for the protection against anoxia/reoxygenation.

## 1. Introduction

The endoplasmic reticulum (ER) is an intracellular organelle that plays an important role in synthesis, folding, modification and degradation, processing and trafficking of proteins [1]. Accumulation of unfolded or misfolded proteins in the lumen of ER results in a cellular stress response named ER stress or unfolded protein response (UPR). To face this ER stress, cells activate UPR adaptive pathways to restore protein homeostasis. Once this mechanism is saturated, unfolded proteins accumulate in the ER lumen and apoptotic pathways are activated. Thus, the balance between adaptative and apoptotic pathways plays a critical role in cell fate along ER stress [2,3]. Under physiological conditions, ER stress sensors (inositol-requiring protein 1 (IRE-1α), protein kinase R (PKR)-like endoplasmic reticulum kinase (PERK) and activating transcription factor 6 (ATF6)) are inactivated by binding to the ER-resident chaperone BiP/GRP78 (immunoglobulin heavy chain binding protein/78 kDa glucose-regulated protein) on the side of the ER lumen [4,5]. In contrast, when cells are exposed to a stress generating proteins misfolding/unfolding, i.e., ER stress, BiP dissociates from these sensors to catch mis/unfolded proteins, hence, leading to the activation of the sensors downstream signaling pathways to tentatively solve this crisis [2]. Activation of this UPR “adaptative” pathway can prevent cell damage by reducing protein translation, increasing ER protein-folding capacity, and ultimately degrading aberrant proteins through ER-associated degradation (ERAD) [4]. Nevertheless, an overwhelming or sustained ER stress can initiate proapoptotic pathways ultimately leading to cell death and tissue damage [4].

Three major UPR downstream pathways exist: IRE-1α–XBP1, PERK–eIF2α–ATF4, and ATF6. XBP1 (X-box-binding protein 1) mRNA, following IRE-1α activation, is spliced and translated into a potent transcriptional activator of UPR (including BiP) and ERAD genes [6]. eIF2α (eukaryotic initiator factor 2α) phosphorylation following PERK activity leads to a general reduction of protein synthesis favouring the resolution of ER stress. Furthermore, phosphorylated eIF2α induces the translocation of specific mRNAs, particularly the one encoding ATF4, which promotes the transcription of UPR genes, including the pro-apoptotic CCAAT-enhancer-binding protein homologous protein (CHOP) [7]. ATF6 is a transcription factor inactivated when bound to BiP. ER stress causes ATF6 cleavage and promotes the expression of several UPR genes including XBP1 and CHOP [8].

Ischemia/reperfusion (I/R) is characterized first by a temporary interruption of blood circulation leading to oxygen and nutrients deprivation in tissue, and then by a sudden reactivation of the circulation involving acute reoxygenation and nutrients supply [9,10]. The kidney is an organ heavily dependent on oxygen supply and is therefore extremely sensitive to I/R, a common risk factor that causes acute kidney injury (AKI) which is associated with dysfunction of other organs [11]. Kidney I/R is unavoidable in transplantation procedures with a warm ischemia period consecutive to the cardiac arrest in organ donation after circulatory death, followed by a period of preservation in optimized cold storage solution currently considered as cold ischemia and finally an acute reperfusion (normothermic reoxygenation) at the time of kidney vascular anastomosis in the recipient [12]. The consecutive shortage in energy and generation of oxidative stress during ischemic [13] and reperfusion steps [14] can damage cells, including macromolecule oxidation and protein homeostasis disturbance (i.e., ER stress), leading finally to cell death [12,15,16]. ER stress is also involved in the development of AKI [3,17] and could be even crucial, when it is persistent, for the transition from AKI to chronic kidney disease [18]. ER stress in AKI promotes renal tubular epithelial cell death, as well as inflammation and autophagy. When I/R or AKI are severe or persistent, cell death and tubular atrophy are associated to secretion of profibrotic factors, hence to renal fibrosis which is extremely deleterious for the organ [19,20]. Interestingly, it has been demonstrated that ER stress is involved in renal fibrosis development and that preventing ER stress, by protecting mitochondria and/or the reticulum membrane, limited I/R injury [21,22]. From a molecular point of view, kidney I/R due to cardiac arrest in mice is associated to increase phosphorylation of both PERK and eIF2α in tubular epithelial cells [16]. These phosphorylations seem localized in the corticomedullary region of the kidney which is highly sensitive to I/R [16], a result confirmed in another model of kidney I/R, i.e., bilateral renal ischemia and reperfusion, where BiP was found activated in the same area [23]. In addition, early after kidney I/R adaptative UPR is induced, characterized by BiP activation, and later, when the stress is sustained, the proapoptotic CHOP is dramatically increased and leads to cell death [18]. Interestingly, UPR stress pathways activation does not seem simultaneous in order to generate an adaptative response depending on intensity and duration of the ER stress. In this manner, it has been demonstrated that the endothelium displayed a time-dependent activation of UPR pathways in response to ER stress when endothelial cells are exposed to cold anoxia and normothermic reoxygenation [24]. In the present study, it could be shown that the ER stress response was regulated in close association with cold ischemia time. Whereas the eIF2α-ATF4 pathway was activated early after the beginning of cold ischemia, its activation decreased when cold ischemia duration was increased. In contrast, the ATF6 pathway was activated later and associated with cell death, while the IRE-1α -XBP1 pathway was activated only at the reoxygenation stage. Both ATF6 and IRE-1α (independently of XBP1) induced CHOP expression during cold ischemia and reoxygenation, and their inhibitions allowed to increase cell survival [24].

Recently a new pharmacological approach identified the polyamine pathway and more particularly the eukaryotic initiation factor 5A (eIF5A) as a pharmacological target related to ischemia tolerance. eIF5A is activated by hypusination, a unique post-translational modification extremely well conserved across evolution [25]. This modification takes place on a specific lysine residue and is carried out from the polyamine spermidine by the successive action of deoxyhypusine synthase (DHPS) and deoxyhypusine hydroxylase (DOHH). eIF5A has been originally characterized as a translation factor involved in protein synthesis initiation, but recently its described functions appear more diversified and complex as it promotes the translation of proteins with particular polyproline residues and participates to the nuclear exports of several mRNAs in cooperation with the exportin complex [26]. Of note, eIF5A exists under two isoforms, A1 ubiquitously expressed, and A2, extensively studied in cancer and displaying alternative functions [27]. We have previously demonstrated that GC7 (N1-guanyl-1,7-diamineoheptane), a spermidine analogue specifically inhibiting DHPS activity and thus hypusination of eIF5A, allows a protection from ischemic events at the cell and organ level [28,29,30]. Transposed to a preclinical model of renal transplantation in pig, GC7 pretreatment of the donor allowed a better functional recovery in the recipient [29,30]. From a metabolic point of view, GC7 decreases oxidative phosphorylation and reversibly reprograms the function and metabolism of kidney cells [31]. This metabolic shift drives cells toward the use of glucose as their single source of energy permitting cells therefore to be oxygen-independent through anaerobic glycolysis [30,31]. In addition, GC7 has been shown to decrease the oxidative stress, a phenomenon well-known to be associated to I/R both in vivo and in vitro, limiting I/R effects [29,30].

Herein we demonstrated that a GC7 preconditioning prevents kidney proximal tubule cell death induced by both cold and warm anoxia, both associated with a strong ER stress. This protective effect is triggered by IRE-1α/XBP1 pathway activation and associated to an inhibition of PERK/eIF2α pathway and an increased expression of the chief chaperone BIP.

## 2. Materials and Methods

### 2.1. Reagents

Drugs, buffer solutions, fetal bovine serum (FBS) and other culture reagents were from Sigma-Aldrich (Merck, Saint-Quentin-Fallaviers, France). N-guanyl-1,7-diaminoheptane (GC7) was synthesized by AtlanChim Pharma (Nantes, France) according to the previously described method [32].

### 2.2. Cell Culture

Renal proximal convoluted tubule cells (PCT) were obtained from primary cultures of murine proximal tubule segments, immortalized with pSV3neo vector and were cultured as previously described [33,34]. Cultures were classically maintained in a 5% CO_2_/95% air water-saturated atmosphere in M1 medium (DMEM/F12, Glutamine, SVF, EGF, T3, dexamethasone, ITS, G418). All experiments were performed the day after cell confluence. For anoxia and hypoxia experiments, cells were maintained at 4 °C or 37 °C in an airtight chamber containing a 100% N^2^ atmosphere or a mix O_2_/N_2_ 1%/99% respectively. Oxygen deprivation was controlled using an OXYBABY^®^ apparatus (Wittgas, WITT, Morsang-sur-Orge, France). Anoxic condition was obtained after 3 h under N_2_ atmosphere when O_2_ levels ≤ 0.1% were detected (Appendix A). Normothermic reoxygenation mimicking reperfusion was performed by placing cells under 5% CO_2_/95% air water-saturated atmosphere at 37 °C.

### 2.3. Cell Survival Analysis

Cell viability was evaluated by using Calcein-AM 1 µM (Invitrogen, Thermofisher Scientific, Illkirch-Graffenstaden, France) and Ethidium homodimer 4 µM (Sigma-Aldrich) to stain live and dead cells respectively. Cells were incubated for 30 min at 37 °C in presence of both dyes and then washed two times with HBSS (w/o Phenol Red) before analysis. Fluorescent micrographs were recorded using an observer D1 microscope (Carl Zeiss, Le Pecq, France) and analyzed with imageJ software. Results are expressed as “survival rate” compared to corresponding normoxia condition.

### 2.4. Protein Analysis

Whole proteins from cells were prepared using TNET lysis buffer (25 mM Tris-Cl (pH 7.4), 100 mM NaCl, 1 mM EDTA, 1% Triton X-100, 0.5% Nonidet P40, 1× protease inhibitor cocktail and 1× Phosphostop mix (Roche Diagnostics, Meylan, France)). Crude lysate was centrifuged at 10,000× *g* (30 min, 4 °C). Supernatants containing soluble proteins were preserved for analysis. For ATF6 expression analysis, nuclear proteins were prepared as previously described [35], and normalized using unspecific band revealed by Amido Black staining of the corresponding Western blot.

Protein concentration was evaluated by BCA assay (PIERCE, Thermofisher Scientific) and blotted using SDS-PAGE basic protocol and Mini-PROTEAN^®^ TGX™ Precast Protein Gels (BioRad, Marnes-la-Coquette, France). Primary antibody incubation was performed overnight at 4 °C (1:1000, bovine serum albumin 3%, anti-PDI, anti-BIP, anti-eIF2α and anti-phospho-eIF2α are from Cell Signaling Technology (Ozyme, Saint-Cyr-l’Ecole, France), anti-β-tubulin and anti-ATF6 is from Abcam, Cambridge, UK) and then with an adequate HRP-conjugated secondary antibody (Jackson ImmunoResearch, Ely, CB7, United Kingdom) (30 min, bovine serum albumin 3%, 1:5000, RT). Detection was performed using Immobilon Western Chemiluminescent HRP Substrate (Millipore, Merck) and Fuji apparatus. Band intensities were evaluated using Fiji Software (Fiji 2.9.0, Opensource, https://fiji.sc, accessed on 4 December 2022.).

### 2.5. Immunofluorescence

Cells were fixed with 4% paraformaldehyde for 10 min, permeabilized with 0.1% Triton X-100/BSA 3% for 10 min, and then sequentially incubated with primary antibody overnight at 4 °C (anti-CHOP, Cell Signaling technology #2895, dilution 1:100) and with the relevant secondary antibody coupled to Alexa-594 (Invitrogen, dilution 1:500) for 30min at RT. Cells were finally mounted and visualized with an inverted epifluorescence Axiovert microscope (Carl Zeiss, Le Pecq, France) under oil immersion and photographs were acquired and processed with Zen software (Carl Zeiss).

### 2.6. mRNA Analysis

Procedures follow MIQE recommendations [36]. Total RNA was extracted using TRIzol (Invitrogen, Thermofisher Scientific) according to the manufacturer’s instructions. In addition, tissues were solubilized using Precellys tissue homogenizer in TRIzol reagent and using CK14 beads. Reverse transcription-polymerase chain reaction (RT-PCR) was performed using M-MLV-RT (Promega, Charbonnières-les-Bains, France). SYBR qPCR premix Ex TaqII from Takara (Ozyme) was used for quantitative PCR (qPCR), and assays were run on a StepOne Plus ABI real-time PCR machine (PerkinElmer Life and Analytical Sciences, Boston, MA, USA). Expression of selected genes was normalized to that of *gapdh* and *rplp0* housekeeping genes, and then quantified using the comparative-ΔCt method. Primer sequences are available upon request.

### 2.7. Statistical Analysis

Data were analyzed by GraphPad Prism 6 software and using ANOVA followed by Tukey’s post hoc test to assess statistical differences between multiple experimental groups. Differences were considered statistically significant with *p* < 0.01. Data are displayed as scatter plot of independent values and group mean values ± SD.

## 3. Results

### 3.1. GC7 Preconditioning Prevents Anoxia- and Tunicamycin-Induced Cell Death

We previously demonstrated that inhibition of eIF5A hypusination by GC7 prevented anoxia-induced cell death when applied for 24 h before stress. Here, we modified our pretreatment protocol to highlight the potential preconditioning effect of GC7 in a different time frame. For this, confluent kidney cells, originating from the Proximal Convoluted Tubule (PCT cells [33]), were treated with 30 µM GC7 for only 8 h, washed and maintained in culture medium without GC7 for an additional period of 24 h before being exposed to stressful conditions (Figure 1A). We observed that this preconditioning protocol was sufficient to protect cells from warm anoxia- induced cell death (24 h, 37 °C, ≤0.1% O_2_) as assessed by the live/dead fluorescent labelling assay (Figure 1B). Indeed, up to 90 % of PCT cells died after 24 h of anoxia as previously described, while less than 20 % of cells died when they were pretreated with GC7 (Figure 1C). To further mimic in vivo transplantation conditions (cold ischemia/reperfusion process), we exposed the cells to cold anoxia (16 h, 4 °C, ≤0.1% O_2_) followed or not by a normothermic reoxygenation step (2 h, 37 °C, 20% O_2_). We chose 16 h of anoxia because increasing the time of cold anoxia led to maximal death of untreated cells and thus did not allow to study the impact of reoxygenation. Untreated PCT cells showed a death rate of approximately 60% after cold anoxia and 80% after reoxygenation (Figure 1C). Mortality was completely prevented when cells were pretreated with GC7 24 h before.

GC7 treatment has been shown to induce a metabolic shift toward anaerobic glycolysis and limit oxidative stress generation, two effects protecting cells from O_2_ deprivation [30,31]. Nevertheless, several studies demonstrated the key role of ER stress in the alteration of cell function and survival during ischemia/reperfusion [24]. To determine if GC7 was able to protect cells from cell death induced by UPR activation, we treated cells with 10 µg/mL of tunicamycin which is a well-known inductor of ER stress. As shown in Figure 1, cells pretreated with GC7 were protected from cell death induced by a 24 h tunicamycin treatment. To define the part of ER stress in cell death induced by warm and cold anoxia, we also analysed *chop* mRNA expression and CHOP nuclear translocation which is the ultimate event linking ER stress to cell death. *Chop* mRNA expression was increased after either warm or cold anoxia and reoxygenation in control cells but not in cells pretreated with GC7 (Figure 2A). Concomitantly, PCT cells exposed to warm or cold anoxia and reoxygenation exhibited a strong positive nuclear staining for CHOP (Figure 2B). This staining was absent in cells pretreated with GC7 prior the anoxic exposure. These interesting results suggested a new link between the modulation of ER stress and the protective effect of GC7. They also prompted us to thoroughly analyzed the expression and activation of the different ER stress pathways components in PCT cells preconditioned with GC7 and exposed to anoxic conditions.

### 3.2. GC7 Modulates Differentially ER Stress Pathway Activation during Anoxia

To investigate the activation of the different ER stress pathways, we first assessed *xbp1* mRNA splicing and the expression of two of its target genes (*bip* and *edem*) as indicator of IRE-1α pathway activation. Warm anoxia, causing dramatic mortality after 24 h, induced a strong splicing of *xbp1* correlated with increased expression of *bip* (Figure 3A). GC7 pretreatment did not modify *xbp1* splicing and *bip* mRNA expression under warm anoxia (Figure 3A) but strongly increased BiP protein level (Figure 3B). Otherwise, *edem* mRNA expression was not modified under warm anoxia independently of GC7 pretreatment (Figure 3A). Interestingly, when cells were maintained under normoxia, pretreatment with GC7 induced a slight but significant increase in *xbp1* splicing, in *edem* and *bip* mRNA levels as well as in BiP protein level (Figure 3A,B). To analyze PERK pathway, we evaluated eIF2α phosphorylation which is due to PERK activation. Warm anoxia induced eIF2α phosphorylation, which was not the case when cells were pretreated with GC7 (Figure 3B). Nuclear cleaved ATF6, which is an indicator of ATF6 pathway activity, was strongly decreased under warm anoxia independently of GC7 pretreatment (Figure 3B). Finally, Protein Disulfide Isomerase (PDI) level, a chaperone protein rectifying incorrect disulfide bonding involved in ER stress resolution, were equivalent in untreated and GC7 treated cells both under normoxia and warm anoxia (Figure 3B).

To understand the protective mechanism provided by GC7 treatment, we decided to expose PCT cells to cold anoxia for 4 or 16 h followed or not by 2 h of warm reoxygenation. Of note, in contrast to 16 h of cold anoxia, 4 h under this condition did not induce cell death or CHOP nuclear translocation even after normothermic reoxygenation (data not shown). Under control condition, 4 h of anoxia followed or not by normothermic reoxygenation did not affect mRNA levels of ER stress pathways but induced ATF6 nuclear localization. Longer anoxia (16 h) increased *xbp1* splicing and eIF2α phosphorylation (Figure 4A,B). Interestingly, no clear modulation of ER stress pathways was found after 16 h of cold anoxia and 2 h of reoxygenation, suggesting normalization of protein folding and translation function in cells. As expected, PDI levels remained unchanged across all tested conditions (Figure 4B). Moreover, following 4 and 16 h of anoxia, GC7 pre-treatment increased *xbp1* splicing correlated with an increased expression of *edem* and *bip* mRNAs (Figure 4A). These expressions were further increased after 2 h of reoxygenation, certainly due to sustained activity of the spliced XBP1 protein despite the decrease in spliced mRNA level. *bip* mRNA over-expression was confirmed at the protein level, with an increase in the level of BIP whatever the time of cold anoxia and reoxygenation (Figure 4B), while the PDI level did not change after GC7 preconditioning. Nuclear ATF6 levels and eIF2α phosphorylation were decreased after 4 h of cold anoxia and after 16 h for eIF2α when cells have been pretreated with GC7 (Figure 4B).

Similar results were obtained when the cells were exposed to mild hypoxia (1% O_2_, 37 °C) for 24 h. Indeed, GC7 preconditioning resulted in increased *xbp1* splicing concomitantly with increased BIP mRNA and protein expressions (Appendix A). It is interesting to note that, mild hypoxia alone does not lead to cell death, suggesting a potential broader/conserved protective mechanism.

Together, these results demonstrate that GC7 preconditioning increased IRE-1α pathway activity during warm and cold anoxia but inhibited PERK and ATF6 pathways during cold anoxia and normothermic reoxygenation.

### 3.3. Involvement of IRE-1α Pathway in GC7 Protective Effect

To determine the importance of the IRE-1α pathway in the protection conferred by the GC7 preconditioning, we used 4µ8c (1 µM), a known inhibitor of IRE-1α to prevent *xbp1* splicing (Appendix A). 4µ8c treatment did not change control cells viability to either cold or warm anoxia but led to a partial inhibition of the protective effect of GC7 against warm anoxia (Figure 5A) and a complete inhibition against cold anoxia (Figure 5B). Interestingly, under normoxia, 4µ8c treatment did not show any toxic effect on PCT cells in presence or absence of GC7 (Figure 5).

An mRNA analysis demonstrated that 4µ8c treatment at the time of both warm and cold anoxia resulted in an expected inhibition of *xbp1* splicing (Figure 6), particularly visible with the decrease in spliced *xbp1* without affecting the level of unspliced *xbp1* mRNA (Appendix A). Concomitantly, as *bip* and *edem* expressions are partly controlled by spliced XBP1, we can observe a decrease in mRNA expressions, but only partial following 4µ8c treatment (Figure 6).

Inhibition of XBP1 splicing as well as the decrease in *bip* mRNA expression could be associated with the loss of the GC7 protective effect. Analysis of BiP protein level did not confirm this hypothesis. Indeed, while BiP level was increased in PCT cells pretreated with GC7, we did not observe a decrease of BiP when cells were treated with 4µ8c during anoxia (Figure 7). Furthermore, we have previously shown that GC7 pretreatment limited *chop* mRNA expression during warm and cold anoxia (Figure 2A) and *chop* expression is partly controlled by spliced XBP1. Surprisingly, this inhibition of *chop* mRNA expression in GC7 pretreated cells was not prevented by 4µ8c treatment (Figure 6).

### 3.4. Impact of GC7 Preconditioning on ER Stress Pathway

A preconditioning treatment is thought to prepare cells to resist to future stress by inducing a small stress response, according to the hormesis concept. Thus, we analyzed PCT cells at the end of the 8 h of GC7 treatment and 16 h later. At the end of treatment, GC7 did not induce by itself an activation of ER stress pathways (Figure 8A,B). In contrast, 16 h later PCT cells pretreated with GC7 displayed an increase in *bip* and *edem* mRNA levels which were related to a slight increase in *xbp1* splicing (Figure 8A). While phosphorylation of eIF2α decreased without reaching significance 16 h after GC7 pretreatment, a significant increase in the amount of BiP protein was found (Figure 8B).

## 4. Discussion

Warm and cold ischemia as well as reperfusion lead to various organ/cells damages [13,14] due to energy shortage, oxidative stress and ER stress [15,16]. Numerous studies have described the benefit of many compounds when used as a pre- or per-conditioning I/R treatment, notably GC7 [37]. This compound is a specific inhibitor of DHPS and thus of eIF5A hypusination, and has been characterized as a valuable in vivo protective agent against renal I/R in rat and pig [29,30]. From a mechanistic point of view, preconditioning with GC7 leads to metabolic changes as it significantly decreases mitochondrial oxidative phosphorylation and promotes anaerobic glycolysis, which is a mechanism of adaptation and response to oxygen deprivation. This allows cells to survive under anoxic conditions for more than 24 h, with the cells completely returning to their original metabolic status after a 48 h recovery period under normoxic conditions [30,31]. In addition, preconditioning with GC7 lowers oxidative stress under anoxic and reoxygenation conditions, although it has not been characterized whether this is a cause or a consequence of the GC7-induced protection [30]. Similar results were found in an animal model of brain-dead donor pigs, in which preconditioning with GC7 decreased oxidative stress production in the graft [29]. In conclusion, GC7 preconditioning prevents energy shortage and limits oxidative stress in kidney submitted to I/R, which participates to improve graft function recovery. Nevertheless, GC7 preconditioning impact on ER stress remained to be defined as this stress has a pivotal role in graft outcome.

Kidney I/R in mice activates ER stress pathways, first as an adaptative response early after ischemia, and later when ischemia is sustained to a proapoptotic response [18]. These activations seem mainly localized in tubular epithelial cells, particularly within the corticomedullary junction which is a region highly sensitive to oxygen variation [16,23]. ER stress is highly deleterious and central in the sensitivity of kidney to I/R, and modulation of ER stress protected partly from I/R injury. Interestingly, several studies linked ER stress activation to the development of renal fibrosis after I/R [21,22]. Hence, we previously demonstrated that GC7 preconditioning of donor pigs limited drastically the development of renal fibrosis in recipients several months after the graft [29,30], potentially linking GC7 treatment to ER stress prevention. Herein, we demonstrated that PCT cells pre-treated with GC7 were able to resist to either warm or cold anoxia, and that resistance was linked to a decreased in CHOP nuclear translocation, known to induce an apoptotic program. Similarly, we found a decrease of PERK pathway activity characterized by a reduced phosphorylation of eIF2α. Nevertheless, GC7 did not inhibit all UPR pathways as we found an increased *xbp1* splicing and BiP expression. This demonstrated a differential impact of GC7 treatment on UPR stress pathways, perhaps highlighting a positive effect on the adaptive response concomitantly with an inhibition of proapoptotic ER stress pathways.

It is well-known that ER stress pathways activation is not simultaneous and can lead to adaptative or proapoptotic responses. This is especially true for endothelial cells during I/R episodes, as they display a sequential activation of UPR pathways depending on the duration of anoxia and leading first to adaptative response and then to cell death [24]. Using the same approaches on PCT cells, we showed that, while IRE-1α is activated early and sustained along cold anoxia, ATF6 was only activated in the first hours while PERK pathway was only activated during sustained anoxia. As expected, CHOP nuclear translocation happened after 16 h of cold anoxia but not for shorter exposure times (4 h anoxia, not shown), correlating anoxia duration and cell death rate. This response to anoxia of tubular epithelial cells is sequential but different from that observed in endothelial cells [24], demonstrating the multiplicity and the complexity of the UPR response in different cells/tissues facing adverse conditions.

GC7 preconditioning led to an increase of *xbp1* splicing and BiP expression, the first certainly leading to the second. We assume that increased BiP levels in GC7 pretreated cells allow a higher capacity to chaperone proteins for their folding. This could increase the cell’s tolerance threshold to excessive folding demand at the time of ER stress, here induced by anoxia, and consequently limiting the activation of UPR downstream pathways, PERK–eIF2α–ATF4 and ATF6. As no nuclear ATF6 was found after 16 h (cold) or 24 h (warm) of anoxia, we can postulate that the PERK pathway controls CHOP expression in PCT cells. Thus, the survival level obtained in GC7 pretreated cells could be due ultimately to the inhibition of this pathway characterized in our work by the inhibition of eIF2α phosphorylation under anoxia. Altogether, these results suggest that GC7 treatment induces *xbp1* splicing which in turn increases BiP levels allowing to maintain inactive the PERK pathway at the time of stress. Increased splicing of *xbp1* was a necessary step to trigger the GC7 protective effect, as splicing inhibition by the IRE-1α inhibitor 4µ8c reversed GC7 mediated protection. Intriguingly, while we correlated this to reduce *bip* mRNA levels, we did not confirm this decrease at the protein level. As 4µ8c was added on cells only during anoxia, we can suppose that the increase in BiP protein levels found under anoxia was more related to the effect of GC7 preconditioning before stress which potentially enhances the tolerance of cells to stress. In contrast, during anoxia the increase in XBP1 splicing seemed related to another stress tolerance pathway independent of BiP and involving one of the other actors of UPR not identified in our work.

We did not link the loss of GC7 protective effect under 4µ8c treatment to an increase in *chop* expression. This was not completely surprising because *chop* mRNA expression is mainly controlled by ATF4 and ATF6 pathways instead of by IRE-1α but it raises the question about the loss of the protective effect which could be unrelated to the CHOP pathway. Interestingly, in insulinoma cells, GC7 treatment protected cells from ER stress-induced death by decreasing CHOP level and caspase-3 cleavage, without affecting *chop* expression. In this situation, it was proposed that eIF5A hypusination was required for CHOP translation and thus that GC7 inhibited it [38]. As CHOP does not contain polyproline residue and has not been identified as a translational product of hypusinated eIF5A [39,40,41,42], we can suspect another role. Here, using a preconditioning treatment, we report a similar protective effect as observed in insulinoma cells, therefore excluding a direct and acute effect of GC7 on CHOP translation. Therefore, the underlying mechanism of the protective effect against ER stress-induced cell death conferred by GC7 treatment remains to be discovered.

Nevertheless, we clearly demonstrate that GC7 preconditioning delayed the activation of UPR when cells were exposed to a stress. This could be associated with an alteration of protein translation due to eIF5A hypusination inhibition by GC7 as hypusinated eIF5A is globally involved in translation, from translation initiation to the termination of multiple proteins [43]. Thus, several hypotheses are still relevant to explain the effect of GC7 on ER stress: (1) a decrease in protein synthesis which could limit ER stress amplitude; (2) an inhibition of translation of specific proteins involved in or modulating UPR; (3) an indirect effect involving the role of eIF5A in the translation of proteins involved in mitochondria activity and oxidative stress, both involved in ER stress activation [39].

A deeper analysis of the multiple pathways involved in ER stress and/or cell apoptosis is required to clearly characterize the protective effect of eIF5A hypusination inhibition under anoxia. Nevertheless, we demonstrated clearly that GC7 treatment modulates UPR by activating IRE-1α/XBP1 pathway while inhibiting PERK/eIF2α, concomitantly to preventing CHOP nuclear translocation. Together these results demonstrated a new protective effect conferred by GC7 against ER stress-induced cell death under anoxic environment. Knowing that ER stress has been linked to the development of renal fibrosis [19,20,21,22], the in vitro effect described here can be related to the anti-fibrotic action of GC7 during kidney transplantation [29,30]. Also, we must now transpose these results in more physiological situations, such as in an in vivo kidney model of warm and cold ischemia followed by reperfusion.

## Figures and Tables

**Figure 1 cells-12-00409-f001:**
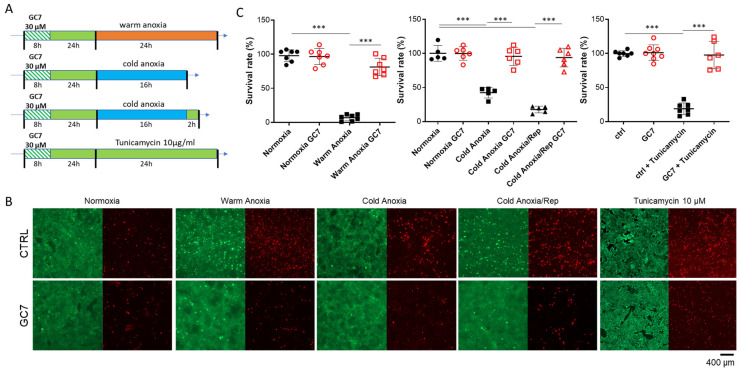
GC7 pretreatment protects PCT cells from anoxia- and tunicamycin-induced cell death. (**A**) PCT cells pretreated for 8 h either with a vehicle or with 30 µM GC7 and 24 h later, exposed to either normoxia, warm anoxia (≤0.1% O_2_ at 37 °C for 24 h) or cold anoxia (≤0.1% O_2_ at 4 °C for 16 h) followed or not by a normothermic reoxygenation step (“Rep”, 20% O_2_ at 37 °C for 2 h), or treated with tunicamycin 10 µg/mL. At the end of the experiment, cell viability was evaluated using a live/dead fluorescence assay. (**B**) Representative images of each condition with live (green, calcein-FITC) and dead (read, ethidium bromide homodimer) cells and (**C**) Corresponding survival rate (live/dead cells). Scatter plot representation of individual values along with mean ± SD for each condition, n = 5–7. *** = *p* ≤ 0.0001.

**Figure 2 cells-12-00409-f002:**
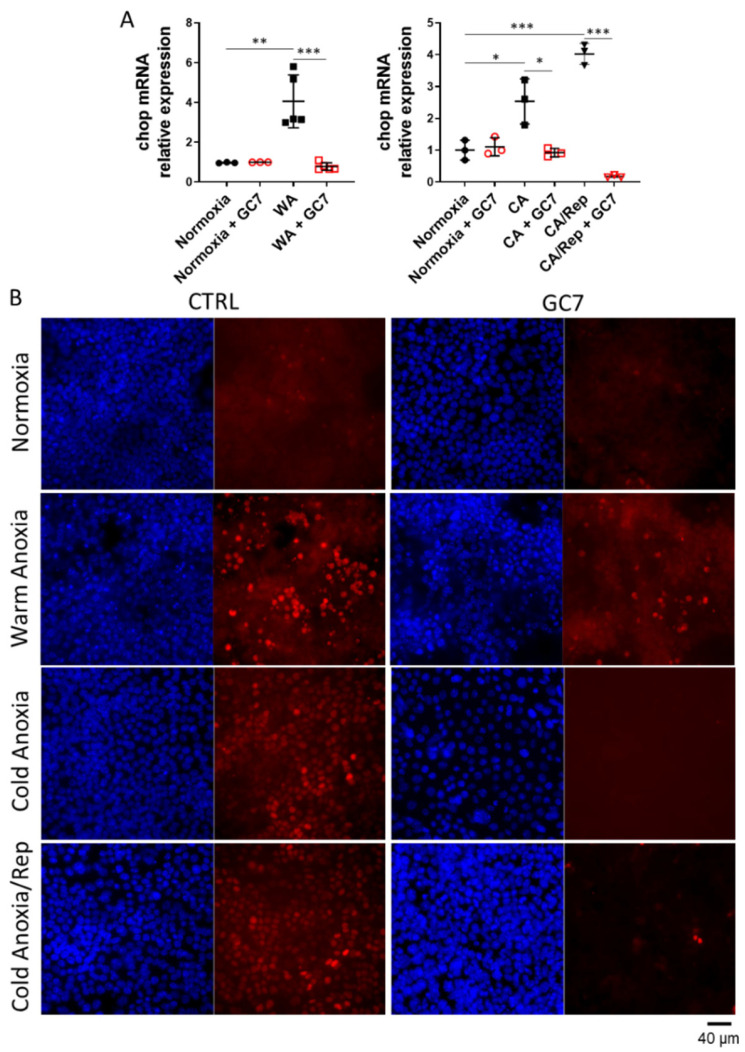
GC7 pretreatment prevents the rise in CHOP expression and its nuclear translocation following anoxia and reoxygenation. PCT cells were pre-treated for 8 h either with a vehicle or with 30 µM GC7 and 24 h later, exposed to either normoxia, warm anoxia (≤0.1% O_2_ at 37 °C for 24 h), or cold anoxia (≤0.1% O_2_ at 4 °C for 16 h) followed or not by a normothermic reoxygenation step (“Rep”, 20% O_2_ at 37 °C for 2 h). At the end of the experiment cells were either (**A**) lysed for mRNA expression analysis or (**B**) fixed and processed for CHOP immunodetection. (**A**) Scatter plot representation of individual values along with mean ± SD for each condition of chop mRNA n = 3–5, * = *p* ≤ 0.01; ** = *p* ≤ 0.001; *** = *p* ≤ 0.0001. (**B**) Representative images of CHOP immunostaining (red) and nuclei counterstained with DAPI (blue).

**Figure 3 cells-12-00409-f003:**
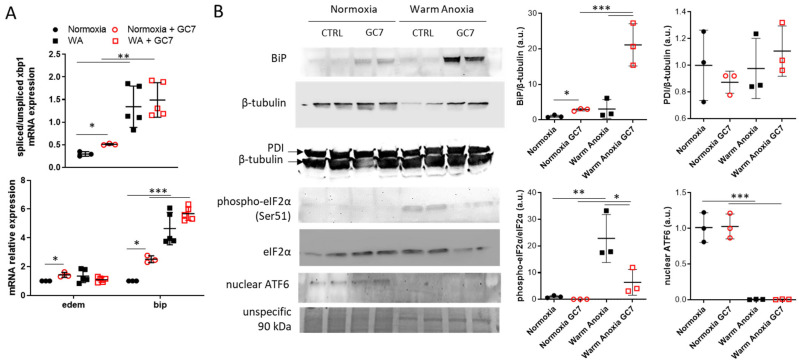
GC7 modulates ER stress response induced by anoxia. PCT cells pretreated for 8 h either with a vehicle or with 30 µM GC7 and 24 h later, exposed to either normoxia or warm anoxia (≤0.1% O_2_ at 37 °C for 24 h). At the end of the experiment cells were either lysed for (**A**) mRNA or (**B**) protein expression analysis. (**A**) Scatter plot representation of individual values along with mean ± SD (n = 3–5) for the ratio of spliced/unspliced *xbp1* mRNA and the relative expressions of *edem* and *bip*. (**B**) Representative western blots of BiP, PDI, eIF2α (total and phosphorylated on Ser51 site) and ATF6 protein levels and the scatter plot representation of normalized individual values along with mean ± SD, n = 3. * = *p* ≤ 0.01; ** = *p* ≤ 0.001; *** = *p* ≤ 0.0001.

**Figure 4 cells-12-00409-f004:**
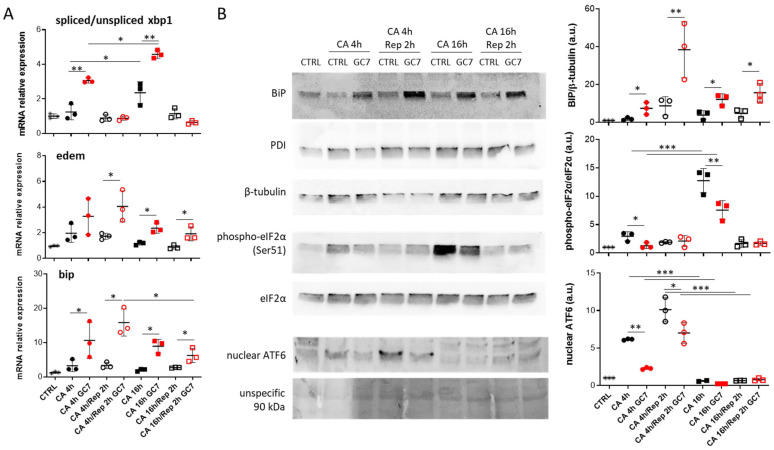
GC7 modulates ER stress response induced by cold anoxia and normothermic reoxygenation. PCT cells pretreated for 8 h either with a vehicle or with 30 µM GC7 and 24 hours later, exposed to either cold anoxia (CA, ≤0.1% O_2_, 4 °C) for 4 or 16 h and followed or not by normothermic reoxygenation (“Rep”, 37 °C) for 2 h. At the end of the experiment cells were either lysed for (A) mRNA or (**B**) protein expression analysis. (**A**) Scatter plot representation of individual values along with mean ± SD (n = 3) for the ratio of spliced/unspliced *xbp1* mRNA and the relative expressions of *edem* and *bip*. (**B**) Representative western blots of BiP, β-tubulin, PDI, eIF2α (total and phosphorylated on Ser51 site) and ATF6 protein levels and the scatter plot representation of normalized individual along with mean ± SD, n = 3. * = *p* ≤ 0.01; ** = *p* ≤ 0.001; *** = *p* ≤ 0.0001.

**Figure 5 cells-12-00409-f005:**
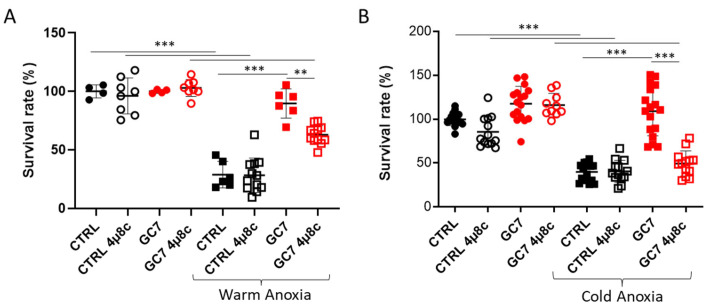
4µ8c inhibition of IRE-1α hampered GC7 protective effect to anoxia. PCT cells pretreated for 8 h either with a vehicle or with 30 µM GC7 and 24 h later exposed to either (**A**) warm (37 °C, 24 h) or (**B**) cold (4 °C, 16 h) anoxia (≤0.1% O_2_), in presence or not of 1 µM of the IRE-1α inhibitor 4µ8c. At the end of the experiment, cell viability was evaluated using a live/dead fluorescence assay. Scatter plot representation of individual values along with mean ± SD for each condition, n = 6–12. ** = *p* ≤ 0.001; *** = *p* ≤ 0.0001.

**Figure 6 cells-12-00409-f006:**
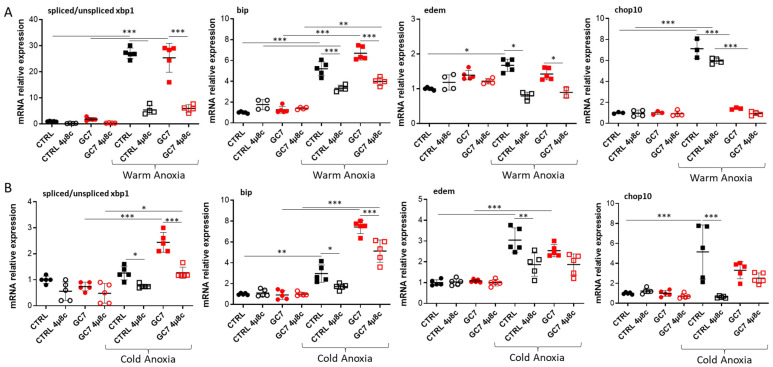
4µ8c effect on the expression of IRE-1α pathway members modulated by GC7. PCT cells pre-treated for 8 h either with a vehicle or with 30 µM GC7 and 24 h later submitted to either (**A**) warm (37 °C, 24 h) or (**B**) cold (4 °C, 16 h) anoxia (≤0.1% O_2_), in presence or not of 1 µM of the IRE-1α inhibitor 4µ8c. At the end of the experiments, cells were either lysed for mRNA expression. Scatter plot representation of individual values along with mean ± SD for the ratio of spliced/unspliced *xbp1* mRNA and the relative expressions of *edem* and *bip*. n = 5, * = *p* ≤ 0.01; ** = *p* ≤ 0.001; *** = *p* ≤ 0.0001.

**Figure 7 cells-12-00409-f007:**
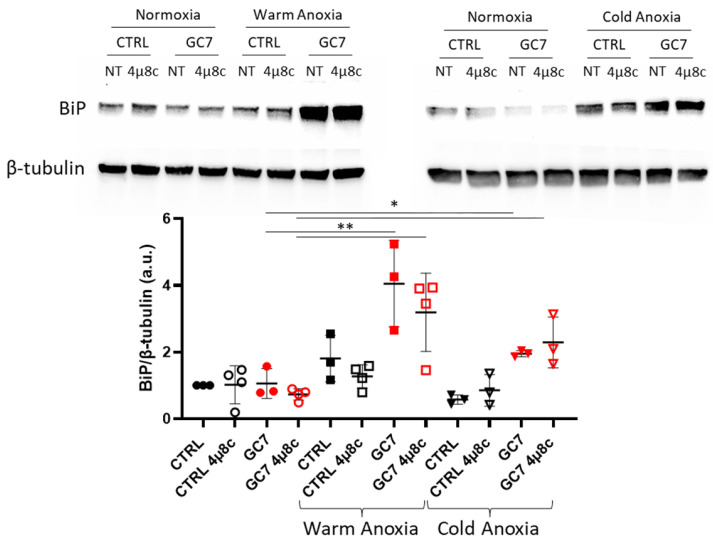
Impact of 4µ8c treatment on BiP protein level. PCT cells pretreated for 8 h either with a vehicle or with 30 µM GC7 and 24 h later exposed to either warm (37 °C, 24 h) or cold (4 °C, 16 h) anoxia (≤0.1% O_2_), in presence or not of 1 µM of the IRE-1α inhibitor 4µ8c. At the end of the experiment cells were lysed for protein expression analysis. Representative western blots of BiP and β-tubulin protein levels. Scatter plot representation of individual values along with mean ± SD for each condition, n = 3–4. * = *p* ≤ 0.01; ** = *p* ≤ 0.001.

**Figure 8 cells-12-00409-f008:**
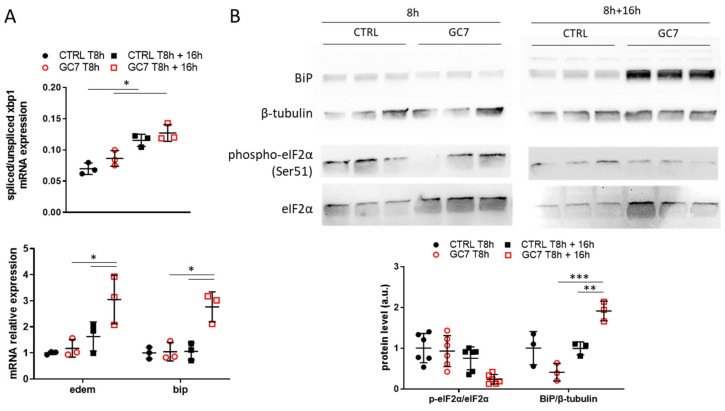
GC7 preconditioning effect on ER stress. PCT cells pretreated for 8 h either with a vehicle or with 30 µM GC7. At the end of treatment or 16 h later cells were either lysed for (**A**) mRNA or (**B**) protein expression analysis. (**A**) Scatter plot representation of individual values along with mean ± SD (n = 3) for the ratio of spliced/unspliced *xbp1* mRNA and the relative expressions of *edem* and *bip*. (**B**) Representative western blots of BiP, β-tubulin and eIF2α (total and phosphorylated on Ser51 site) protein levels and the scatter plot representation of normalized individual values along with mean ± SD, n = 3. * = *p* ≤ 0.01; ** = *p* ≤ 0.001; *** = *p* ≤ 0.0001.

## Data Availability

All data used for analysis have been displayed in the manuscript. Primers were available under request. Images of whole blots displayed in figures and additional blots used for analysis have been uploaded as Appendix A.

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
