# Peer review of "Renal Ischemia Tolerance Mediated by eIF5A Hypusination Inhibition Is Regulated by a Specific Modulation of the Endoplasmic Reticulum Stress"

_cells, 2023, doi:10.3390/cells12030409_

Round 1

Reviewer 1 Report

The manuscript submitted by Melis and collaborators described the role of endoplasmic reticulum (ER) stress in the mechanism involving eIF5A hypusination inhibition during renal ischemia tolerance.

It’s an interesting and nice mechanistic research, with results that could help to develop better pharmacological therapies against anoxia and reperfusion after kidney transplantation.

Looking at the results and the supplementary material, I think the study’s information is clear; only a few minor clarifications would be needed.

Attached three questions that maybe could enforce the paper:

1.     Tunicamycin is a well-known inductor of ER stress, ultimately leading to cell death. Does GC7 protect cells against tunicamycin treatment?

2.     Do the authors have some indications about GC7 impact on ER stress in vivo along with ischemia/reperfusion? 

3.     What is the impact of GC7 preconditioning on ER stress pathway independently of anoxia?

Author Response

First, we would like to thank the two reviewers who took time to assess our manuscript and for the critical and constructive feedback they provided. Following those comments, we revised our manuscript to address their concerns and we hope that this study is now suitable for publication in IJMS. Changes are marked up using the “Track Changes” function within the manuscript and a point-by-point response to the reviewers is provided below in blue.

Reviewer 1.

  1. Tunicamycin is a well-known inductor of ER stress, ultimately leading to cell death. Does GC7 protect cells against tunicamycin treatment?

Thanks to the reviewer for this interesting question. We have decided to not include in the first version of our manuscript this experiment. First to focus our message only on ER stress triggered by anoxia and second because inhibition of glycosylation by tunicamycin is highly different from ER stress induced by a physiological condition. We understand that this point can be of interest for the reader and we have now added these results in Figure 1. As displayed now, GC7 pretreatment protects efficiently cells from tunicamycin-induced cell death. We modified the text accordingly to describe the new added results.

  1. Do the authors have some indications about GC7 impact on ER stress in vivo along with ischemia/reperfusion? 

In a previous study, we have demonstrated that GC7 pretreatment protects kidney function from 15 min ischemia 1h reperfusion induced by a ligature of renal arteria. Analysis of these tissues allowed us to demonstrate that GC7 pretreatment increases BIP expression at the protein level after I/R. Unfortunately, we have not been able to analyze more in-depth these tissues coming from a previous study. In the future, a more extensive analysis of GC7 effect on ER stress will be performed in vivo using perfused rodent kidney models and pig kidney transplant models as previously used by our teams. 

  1. What is the impact of GC7 preconditioning on ER stress pathway independently of anoxia?

We thank the reviewer for this comment which is pointing out an important aspect of GC7 preconditioning. While GC7 preconditioning did not modify ER stress components at the end of the 8 hours treatment, it led to a slight increase of XBP1 splicing, expression of BiP and EDEM mRNA and BiP protein levels 16h after the end of treatment. This seems to demonstrate a preconditioning effect characterized by a little stimulation of UPR pathways, potentially preparing cells to a more deleterious ER stress. These results are now included in the new Figure 8. Manuscript has been accordingly modified to explain and discuss these results.

Reviewer 2 Report

This article discusses the modulation of ER stress during hypoxia and reoxygenation by inhibiting eIF5A hypusination. The researchers worked on a PCT cells model and performed sequences of anoxia, hypoxia, reoxygenation at 37°C or 4°C. Overall the work was well done and the results are robust, however some points need to be clarified.

Overall, the results need to be presented more clearly. Presenting each signaling pathway of ER stress with its own contributors independently would be more readable. Moreover, the microscopy images appear of poor quality. This needs to be corrected.

Page 3, lines 145: the authors talk about simulating reperfusion, but can we really talk about reperfusion on cellular models and even more just after hypoxia or anoxia without nutrient deprivation. Indeed in vivo reperfusion is associated with a resumption of blood flow after an occlusion, and therefore and consecutive to ischemia and not only ischemia. The model needs to be discussed a bit more. It is a cellular model and it is far from what can occur during warm or cold ischemia-reperfusion.

Page 4, line 194: the title of the chapter needs to be changed. The results presented in this part do not highlight any causal relationship between cell death and ER stress.

Page 5, lines 199 to 201: the authors chose to test the effect of GC7 in remote preconditioning. Can they justify this choice and what it can bring in terms of clinical application?

Page 5, line 201 to 210: Authors should justify their choice of anoxia time. Indeed, this differs between warm anoxia and cold anoxia. Moreover, why only study the effects of reoxygenation after cold anoxia?

page 7 line 246 to 259: this chapter is very difficult to read. A reorganization of the figures with the different pathways of ER stress and a rewriting of the results by clearly distinguishing the effects of GC7 in normoxia and anoxia should be carried out.

Page 7: Why are the authors studying BIP expression following CHOP splicing? BIP is an ER stress trigger and not an effector. This has to be clarified.

Page 9, Figure 5: The results shown in Figure 5 should be quantitative and not relative. 

Page 10: The authors hypothesize that the decrease in expression of BIP mRNA by 4p8C is related to the loss of the protective effect of GC7. Why this effect is not visible on the BIP protein? At which level of the pathway could the GC7 acts? Why does it lead to a marked increase in the expression of the BIP protein? Moreover, why the IRE1 alpha inhibitor has no effect on CHOP expression, while it has an effect on mortality?

Author Response

First, we would like to thank the two reviewers who took time to assess our manuscript and for the critical and constructive feedback they provided. Following those comments, we revised our manuscript to address their concerns and we hope that this study is now suitable for publication in IJMS. Changes are marked up using the “Track Changes” function within the manuscript and a point-by-point response to the reviewers is provided below in blue.

Reviewer 2.

- Moreover, the microscopy images appear of poor quality. This needs to be corrected.

Accordingly, we improved the quality of microscopy images in this new version.

- Page 3, lines 145: the authors talk about simulating reperfusion, but can we really talk about reperfusion on cellular models and even more just after hypoxia or anoxia without nutrient deprivation. Indeed in vivo reperfusion is associated with a resumption of blood flow after an occlusion, and therefore and consecutive to ischemia and not only ischemia. The model needs to be discussed a bit more. It is a cellular model and it is far from what can occur during warm or cold ischemia-reperfusion.

We agree that reperfusion is difficult to mimic in vitro. Nevertheless, kidney ischemia/reperfusion is a little bit different that other comparable situation as 1/ tubular cells are less subject to nutrient deprivation due to the tubular environment and the circulating flow is generally lower compared to the general circulation; and 2/ kidney during the transplantation procedure is currently preserved in optimized cold solution which limits in part the nutritive distress (depending of the solution used). Thus, this is different from other ischemic events like for the heart and brain for which the best in vitro procedure consists of Oxygen Glucose Deprivation (OGD) as we have previously used in another work about GC7 ischemic protection (Bourourou JCBFM 2020).

Nevertheless, we are aware that these are imperfect in vitro models. All our results will be more extensively explored as possible in vivo using perfused rodent kidney models and pig kidney transplant models as previously used by our teams.  We have added a sentence in this way at the end of the discussion and we have replaced the term “reperfusion” by “reoxygenation” for all in vitro results. We have better defined cold preservation in the introduction.

- Page 4, line 194: the title of the chapter needs to be changed. The results presented in this part do not highlight any causal relationship between cell death and ER stress.

We agree with the reviewer comment, and we changed the title of this part. As we introduced now in Figure 1 the protective effect of GC7 against Tunicamycin-induced cell death, which is a known inductor of ER stress, we have modified the title for: “GC7 protects cells against anoxia and tunicamycin induced cell death”. In addition, we modified the text accordingly to describe the new added results.

- Page 5, lines 199 to 201: the authors chose to test the effect of GC7 in remote preconditioning. Can they justify this choice and what it can bring in terms of clinical application?

We analyzed the effect of GC7 using preconditioning protocols as already published by our teams previously in rat kidney I/R and in 2 different models of pig kidney transplantation (Melis 2017 JASN, Giraud AJT 2020). This is a relevant procedure in kidney transplantation corresponding to pharmacological donor preconditioning.

The little difference between our different works in vitro is the duration of the treatment (“8h GC7 + 16h relaxation” v/s “24h GC7”). We have shortened the duration and included a time without the molecules before stress to clearly assess the preconditioning effect of GC7 and avoid any effect of the molecule at the time of the stress. In addition, this protocol mimics better the in vivo situation because when we preconditioned animals the day before the experiment (IP or IV injection), we know that GC7 was undetectable in the blood a few hours after the injection. 

- Page 5, line 201 to 210: Authors should justify their choice of anoxia time. Indeed, this differs between warm anoxia and cold anoxia. Moreover, why only study the effects of reoxygenation after cold anoxia?

All our exposure times were determined based on pilot experiments including different time points. Warm anoxia (following circulation arrest in the donor at the time of surgery) in kidney transplantation procedure is not currently followed by reoxygenation. As we wanted to assay in our works originally the protective effect of GC7 on lethality, we have selected 24 hours of warm anoxia exposure. Indeed, shorter duration decreases lethality and therefore potential effect of treatments used, whereas longer exposure does not modify the outcome.

Conversely, cold anoxia (preservation of the kidney in cold nutritive medium without oxygenation) is always followed by reoxygenation in the warm recipient. In the different time points tested, the 16h cold anoxia / 2h reoxygenation is the best condition where we have only partial lethality at the end of cold anoxia, thus allowing study of the reperfusion event. Indeed, increasing the time of cold anoxia led to maximal cell death when cells were not preconditioned with GC7 and thus did not allow to study the impact of reoxygenation.

According to the reviewer’s comments, we have clarified these points in the text (line 214-216).

- page 7 line 246 to 259: this chapter is very difficult to read. A reorganization of the figures with the different pathways of ER stress and a rewriting of the results by clearly distinguishing the effects of GC7 in normoxia and anoxia should be carried out.

We have modified the text according to the reviewer’s comment to group our results following pathways. Nevertheless, we think that our figures (subdivided by techniques) are already sufficiently well organized (instead of subdivided) to delineate UPR pathways.

- Page 7: Why are the authors studying BIP expression following CHOP splicing? BIP is an ER stress trigger and not an effector. This has to be clarified.

As indicated in lines 283-284, BiP is an ER stress trigger for which expression is largely controlled by ER stress pathways and mainly by XBP1 splicing. Indeed, induced expression of BiP by spliced XBP1 allows to prevent cells from an overactivation of UPR pathways, as a negative feedback.

- Page 9, Figure 5: The results shown in Figure 5 should be quantitative and not relative. 

We thank the reviewer for pointing out this error in the legend axis. As for figure 1C, the results correspond of course to the survival rate.

- Page 10: The authors hypothesize that the decrease in expression of BIP mRNA by 4p8C is related to the loss of the protective effect of GC7. Why this effect is not visible on the BIP protein?

We have been highly surprised by these results. We have several hypotheses to explain this discrepancy; one is due to the fact that 4µ8c is applied on cells at the time of stress, and we can suppose that in one way BiP protein levels may be increased by GC7 preconditioning before stress and will enhance the tolerance of cells to stress; and another way, another pathway related to IRE1 participate to stress tolerance during stress and will not involve BiP. We added in the discussion a paragraph about this point (line 523-528).

This is in part supported by the new Figure 8, that we have added in our manuscript to respond to one of the first reviewer’s comments, demonstrating that GC7 preconditioning increased BiP protein level already before the stress period.

- At which level of the pathway could the GC7 acts? Why does it lead to a marked increase in the expression of the BIP protein?

This is a pivotal question highlighted by the reviewer. Unfortunately, our results do not allow to clearly conclude on this point. As we discussed line 543-551, GC7 is a specific inhibitor of eIF5A hypusination which is involved in numerous translation processes. Thus, GC7 treatment could induce a small ER stress (certainly by activation of the IRE1 pathway) preparing the cell to support a more important stress, which is the definition of preconditioning as now suggested by results of figure 8. In our condition, this is characterized by an increase of BiP levels before stress and at the time of stress allowing cells to fold more proteins and at the same time maintain low activation of ER stress downstream pathways.

- Moreover, why the IRE1 alpha inhibitor has no effect on CHOP expression, while it has an effect on mortality?

CHOP is mainly under the control of ATF4 and ATF6 pathways. We suppose that the inhibition of IRE1 was not sufficient to modulate the expression of CHOP under our condition. Of course, this point complexify the system as it demonstrates that the death protection is unrelated to CHOP expression in this condition, and thus point out the involvement of other factors in death protection. We added a sentence in the discussion about this, line 530-532.
